# INJECTING LEARNABLE TABLE FEATURES INTO LLMS

## ABSTRACT

To migrate the remarkable successes of Large Language Models (LLMs), the community has made numerous efforts to extend them to the table reasoning tasks for the widely deployed tabular data. Despite that, in this work, by showing a probing experiment on our proposed StructQA benchmark, we postulate that even the most advanced LLMs (such as GPTs) may still fall short of coping with tabular data. More specifically, the current scheme often simply relies on serializing the tabular data, together with the meta information, then inputting them through the LLMs. We argue that the loss of structural information is the root of this shortcoming. In this work, we further propose TaMo[1], which bears an ideology to treat the **ta**bles **a**s **a**n independent **mo**dality integrated with the text tokens. The resulting model in TaMo is a multimodal framework consisting of a hypergraph neural network as the global table encoder seamlessly integrated with the mainstream LLM. Empirical results on various benchmarking datasets, including HiTab, WikiTQ, WikiSQL, FeTaQA, and StructQA, have demonstrated significant improvements with an average relative gain of **42.65%**.

## 1 INTRODUCTION

Table reasoning, the process of generating task-specific responses based on one or more pre-structured tables rather than unstructured text, has emerged as a key research area. This encompasses various tasks such as table question answering (Pasupat & Liang, 2015), table fact verification (Chen et al., 2019), text-to-SQL (Yu et al., 2018), and predictive tasks (Ye et al., 2024a; Li et al., 2022). Numerous efforts leverage pre-trained language models (LMs) to address these challenges. Classical methods often employ smaller LMs such as BART (Lewis et al., 2020) and T5 (Raffel et al., 2020) to generate answers, often augmented with external retrieval frameworks (Patnaik et al., 2024). However, due to the limited capacity of these smaller models, their methods face challenges in scalability and integration with larger ones.

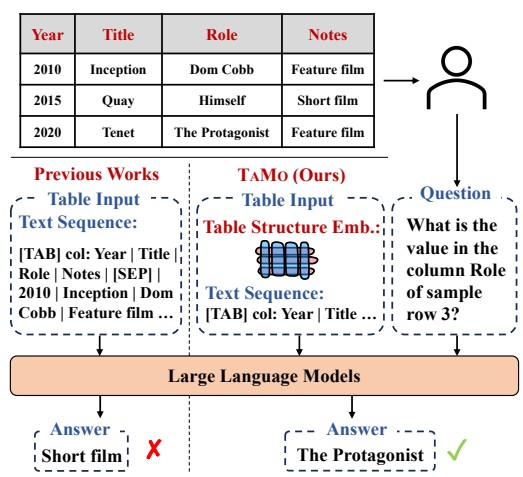

Figure 1: Current tabular LLMs oversimplify tables into text sequences, ignoring structured information and causing poor performance on basic table cell localization tasks. This work is the first to input table structures into LLMs.

With the advent of large language models (LLMs) such as GPT-4 (OpenAI, 2023) and Llama (Touvron et al., 2023), many approaches (Zhang et al., 2024) have attempted to utilize end-to-end LLMs to address table understanding. Despite the effectiveness, a core challenge in this pursuit lies in embedding raw table information within prompts. As shown in Figure 1, an intuitive strategy (Herzig et al., 2020) involves serializing tables into text formats, often using markdown-like markup languages to represent tables, occasionally accompanied by a few examples. However, this method

---

[1]Code and datasets are on `https://anonymous.4open.science/r/HyTaLM-AD2D`

typically suffers from a fundamental problem: ***tables are inherently structured data with permutation invariance***, meaning their semantic content remains unchanged regardless of row or column order. Obviously, the serialized textual formats cannot inherently capture this permutation invariance, making them unsuitable for representing the true nature of tabular data. This concept has been extensively discussed in classical tabular reasoning works (Herzig et al., 2020; Yang et al., 2022), which suggest that a robust table reasoning model should exhibit consistent understanding regardless of such permutations. Yet, this crucial aspect remains underexplored in the context of LLM research.

In this paper, we pose a critical question: ***Can LLMs truly understand tables solely through text-based serialization?*** Unfortunately, our experiments suggest negative. To assess the robustness of LLMs to the permutation-invariance properties of tables, we introduce ***StructQA*** (described in detail in Section 3.2), the first large-scale benchmark designed to evaluate LLMs' comprehension of tabular row and column structures. Specifically, *StructQA* focuses on permutation invariance, assessing whether LLMs can maintain high answer consistency in table question-answering tasks when presented with permuted tables. Surprisingly, as shown in Figure 2, leading LLMs such as Llama2-7B (Touvron et al., 2023), GPT-3.5 (OpenAI, 2022), GPT-4, and TableLlama (Zhang et al., 2023b)—trained explicitly for table tasks—demonstrate poor performance after permutation. Excluding the closed-source GPT-4, their accuracy drops substantially, with answer consistency falling below **40%**. While such identification based on table structures is trivially easy for humans, this phenomenon indicates that ***current LLMs lack a robust grasp and understanding of global table structures***. We hypothesize that serializing tables into text strips away essential structural information, leaving LLMs with limited understanding. When structural perturbations occur, LLMs are prone to hallucinations (Huang et al., 2023) and fragile reasoning.

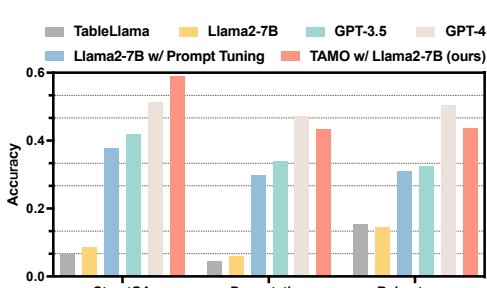

Figure 2: We conducted a probing experiment to evaluate LLMs' table structure understanding using our proposed *StructQA* dataset (detailed in Section 3.1). We tested permutation invariance by randomly permuting rows and columns in the StructQA test set and measured robustness (answer consistency) as the proportion of samples that remain consistent after permutation. TAMO demonstrates superior performance, even competitive with the black-box GPT-4.

**The Imperative of Encoding Tables as an Independent Modality.** To boost robust table reasoning, it is essential for LLMs to explicitly and effectively learn the structural information of tables. However, much like images and audio which contain rich semantic information, tables possess inherent structural nuances that textual serialization fails to represent alone. We draw inspiration from the paradigm of multimodal large language models (MLLM) (Liu et al., 2023; Li et al., 2023). These models learn the semantics of specialized modalities through separate encoding architectures and align different modalities in a unified and more expressive embedding space. This approach, with great success in domains such as graphs (Tang et al., 2024), images (Liu et al., 2023), and audio (Zhang et al., 2023a), innovatively informs our core idea: ***encode tables as an independent modality to integrate their complex relational structures***. By doing so, we can bridge the gap in LLMs' comprehension and achieve a holistic understanding of tables' structure comparable to human cognition through learnable table features.

**Our Approach.** Building on the above intuition, we propose TAMO, a pioneering tabular language model framework to reimagine **Ta**ble representation **a**s **a**n independent **Mo**dality. TAMO leverages theoretically permutation-invariant hypergraph structures to independently capture the intricate relationships and global structures within tabular data. By re-modeling tables as hypergraphs, TAMO effectively combines semantic information of individual table cells (through nodes), with structural information of complex interconnections between cells (through hyper-edges). Harnessing the rich structural information embedded in hypergraphs, TAMO significantly moves beyond traditional sequential text processing on table reasoning. Further, we integrate this hypergraph-based encoding into LLMs through learnable features, achieving dynamic and efficient injection of structural information without tuning the LLM's fixed parameters. This insight offers a more lightweight alignment and

adaptation framework. Consequently, users could avoid the high costs and other potential risks, such as catastrophic forgetting (Zhai et al., 2023), associated with fine-tuning LLMs themselves.

Last but not least, we exhibit extensive empirical validation on four mainstream table reasoning datasets (Hitab (Cheng et al., 2022), WikiTQ (Pasupat & Liang, 2015), WikiSQL (Zhong et al., 2017), and FeTaQA (Nan et al., 2022)) and our proposed *SturactQA* benchmark. TAMo demonstrates substantial performance improvements against previous baselines—up to a **42.65% increase** in average performance. Meanwhile, our methodology validates superior efficacy and broad applicability when integrating hypergraph-encoded tables with diverse LLMs.

**Contributions.** *Position*: Our research represents a revolutionary step in first encoding tables as an independent modality within the LLMs. **Benchmark**: We introduce StructQA, the first open-source benchmark on table structure understanding. Our findings reveal that current LLMs struggle with this human-friendly task. *Methodology*: We explore the hypergraph architecture to capture and model intricate relational structures within varying table formats. This innovative design significantly enhances the table reasoning abilities of LLMs. *Feasibility*: We empirically prove the efficiency of simply and economically training learnable table features to align encoding space with LLMs' semantic manifold.

## 2 METHODOLOGY

For the first time, we treat tables as an independent modality to enhance LLMs' capabilities in table reasoning. In this section, we aim to address the following key questions:

- Section 2.1: **What is table reasoning?**
- Section 2.2: **How to encode the global structural information of the table modality?**
- Section 2.3: **How can table structure and textual information be aligned with LLMs?**

### 2.1 PROBLEM DEFINITION

Following (Wang et al., 2024), table reasoning can be defined as a unified task that acts on samples formatted as triplets $(\mathcal{T}, \mathcal{Q}, \mathcal{A})$. Here, $\mathcal{T}$ represents a pre-structured table containing information clearly organized in rows and columns, with cell types encompassing numerical values, text entries, and dates. $\mathcal{Q} = \{q_1, q_2, ..., q_m\}$ denotes the question or statement related to the table $\mathcal{T}$, typically in a natural language sequence with $m$ tokens. Meanwhile, $\mathcal{A}$ is the expected answer or output of $\mathcal{Q}$, usually simplified into an $n$-tokens sequence $\{a_1, a_2, ..., a_n\}$. Briefly, given the table $\mathcal{T}$ and the question $\mathcal{Q}$, the objective of table reasoning is to predict the corresponding answer $\mathcal{A}$, i.e., $p(\mathcal{A}|\mathcal{T}, \mathcal{Q})$.

### 2.2 HYPERGRAPH-ENHANCED TABULAR ENCODER

A tabular encoder is essential for our multimodal tabular LLMs paradigm. To develop the tabular encoder capable of learning structural information, we first address a fundamental question: "*How to define the structural properties in tabular data?*" As illustrated in Figure 3, we provide the answer based on prior human observations: (i)-most real-world tabular data possess a *hierarchical structure*, with ordinary flat tables being a

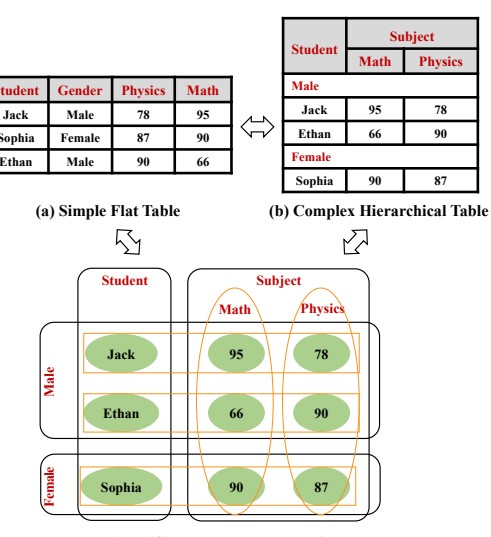

Figure 3: An example of converting arbitrary simple or complex tables into hypergraphs. A simple flat table is a special case of the complex hierarchical table. A hyperedge (e.g., table headers) in the hypergraph is a set of regular nodes. We construct the corresponding hypergraph format according to the hierarchical relationships of the table.

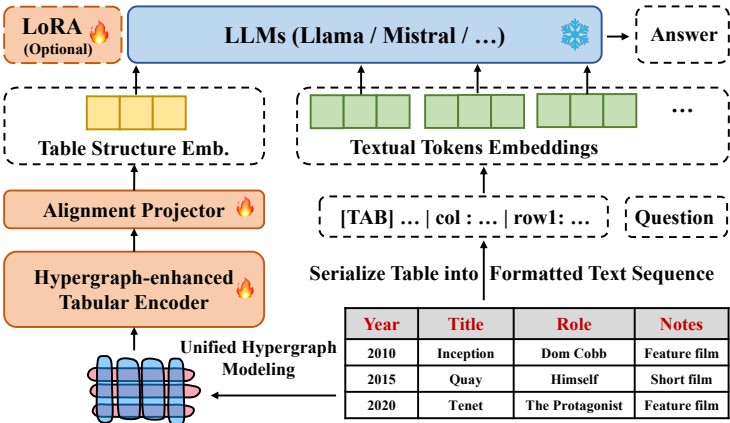

Figure 4: The proposed framework for tabular LLMs, TAMO. Given a table input, the hypergraph-enhanced tabular encoder (Section 2.2) is used to capture the unique structure properties of the tabular modality. Simultaneously, we serialize the original table into a formatted text sequence. Finally, we input both the table structure and textual embeddings into LLMs, generating answers using the next token prediction paradigm. LoRA is optional.

special case of this hierarchy; (ii)-cells within each hierarchy and hierarchies at the same level exhibit *permutation invariance*. For example, arbitrarily swapping rows or columns in a table does not distort its original meaning. This implies that learning the relationships between table cells should not be pairwise but rather set-based. Building on the inherent hierarchical structure of tables, we introduce the **hypergraph** (Yadati et al., 2019) architecture to model tabular data. This approach incorporates both *high-order hierarchical structure* and *permutation invariance* as inductive biases, enabling the precise modeling of complex structural properties in tabular data. For the first time, it allows us to successfully model all types of tables, from simple flat tables to complex hierarchical forms (Cheng et al., 2022).

We re-construct the structure of tabular data via hypergraph. Specifically, a hypergraph $\mathcal{G} = (\mathcal{V}, \mathcal{E})$ consists of a set of nodes $\mathcal{V}$ and hyperedges $\mathcal{E}$. Each hyperedge $e \in \mathcal{E}$ is a subset of $\mathcal{V}$, i.e., $e \subseteq \mathcal{V}$. For a table $\mathcal{T}$, we represent each leaf cell, defined as a cell that does not contain any other cells within the hierarchy, as a node $v \in \mathcal{V}$ and each branch cell, defined as a cell that contains other cells within the hierarchy, as a hyperedge $e \in \mathcal{E}$. Each hyperedge $e$ consists of nodes that belong to its hierarchical level. For example, in a simple flat table, each table cell is a node, and each column or row is a hyperedge encompassing all nodes within that column or row. Under this modeling, altering rows or columns maintains a consistent graph structure (both nodes and edges), effectively reflecting the *permutation invariance* of tables.

Furthermore, to learn the information propagation between nodes and hyperedges in the hypergraph, we construct the **hypergraph-enhanced tabular encoder** with two types of multiset functions (Chien et al., 2021). In this way, we aim to capture *higher-order hierarchical structures* in hypergraphs effectively. The multiset function is defined as a function that satisfies the *permutation invariance* property. Inspired by (Chen et al., 2024), we combine the two types of multiset functions serially, as shown in Eqa.1 and Eqa.2. Specifically, every layer of the tabular encoder we construct includes two parts. The first part is a multiset function that aggregates node information to update hyperedge representations:

$$\mathbf{x}_e^{t+1} = Fusion(\mathbf{x}_e^t, Multiset_1(\{\mathbf{x}_v^t \mid v \in e\})), \tag{1}$$

where $t$ refers to the current layer number; $\mathbf{x}_v$ is the embedding of the node $v$; $\mathbf{x}_e$ is the embedding of the hyperedge $e$; the *fusion* layer is employed to integrate hyperedge information from the last layers, typically utilizing a multilayer perceptron (MLP) network.

The second part is another multiset function that aggregates hyperedge information to update node representations:

$$\mathbf{x}_v^{t+1} = Multiset_2(\{\mathbf{x}_e^{t+1} \mid v \in e\}). \tag{2}$$

Finally, we use the Set Transformer (Lee et al., 2019) to parameterize these multiset functions for learning. Each set attention block is defined as:

$$Multiset(\mathbf{X}) = LayerNorm(\mathbf{H} + rFF(\mathbf{H})),$$
$$H = LayerNorm(\mathbf{X} + MultiHead(\mathbf{S}, \mathbf{X}, \mathbf{X})), \tag{3}$$

where $\mathbf{S}$ is a trainable parameter vector; $rFF$ is the row-wise feedforward layer; $LayerNorm$ is layer normalization (Ba et al., 2016); $MultiHead$ is the multi-head attention mechanism (Vaswani et al., 2017). By facilitating the mutual propagation of information between nodes and hyperedges, the model effectively learns the complex hierarchical relationships among table cells thus outputting learnable table features.

## 2.3 A Modality Interface for Integrating Table Structure Representations with LLMs

Most LLMs (Meta, 2024; Jiang et al., 2023a; OpenAI, 2022; 2023) are pre-trained on large-scale unlabeled corpora in an *autoregressive* manner, thereby learning rich linguistic structures and patterns. To maximize the utilization of LLMs' powerful text understanding and reasoning capabilities for table reasoning tasks, we design a fully *autoregressive* interface to integrate structure representations from the tabular modality with LLMs for table reasoning tasks. The overall framework of our proposed TAMO is shown in Figure 4. We inject the structure representations learned by the hypergraph-enhanced tabular encoder in Section 2.2 into the LLMs in a manner similar to the soft prompt (Lester et al., 2021). *This allows the LLMs to globally perceive the structural information of the tabular data before reading the textual information*, thereby enhancing their understanding and reasoning abilities regarding tabular tasks.

**Aligning Table Structure Representations to LLM Semantic Space.** Assuming the node representations obtained through the tabular encoder are $\hat{\mathbf{X}}_\mathcal{V} = \{\hat{\mathbf{x}}_v | v \in \mathcal{V}\} \in \mathbb{R}^{|\mathcal{V}| \times d_g}$, and the hyperedge representations are $\hat{\mathbf{X}}_\mathcal{E} = \{\hat{\mathbf{x}}_e | e \in \mathcal{E}\} \in \mathbb{R}^{|\mathcal{E}| \times d_g}$. $d_g$ is the hidden dimension of the tabular encoder. We use a multilayer perceptron (MLP) network to learn the transformation of table structure representations $\mathbf{X}_{st}$ into the semantic space:

$$\mathbf{X}_{st} = MLP(Pooling(\hat{\mathbf{X}}_\mathcal{V}, \hat{\mathbf{X}}_\mathcal{E})) \in \mathbb{R}^{d_l}, \tag{4}$$

where *pooling* is an information aggregation function for nodes and hyperedges, set up as *mean pooling* in our experiment; $d_l$ is the hidden dimension of LLMs.

**Generating Answers based on both Tabular and Textual Modality Information.** Following previous works (Zhang et al., 2023b; Wang et al., 2024; Herzig et al., 2020), we serialize tabular data into formatted text sequences and obtain the text embeddings of tabular data $\mathbf{X}_{tt} \in \mathbb{R}^{L_s \times d_l}$ through the LLMs' embedding layer. $L_s$ is the length of text sequences. For questions in natural language form, we obtain the corresponding question tokens $\mathbf{X}_{qt} \in \mathbb{R}^{L_q \times d_l}$ through the embedding layer similarly. $L_q$ is the length of question sequences. The final answer is generated following the next token prediction paradigm:

$$p(\mathcal{A}|\mathcal{T}, \mathcal{Q}) = \prod_i^n p(a_i \mid \mathbf{X}_{st}, \mathbf{X}_{tt}, \mathbf{X}_{qt}, a_{j<i}), \tag{5}$$

where $n$ is the number of answer tokens $\mathcal{A} = \{a_1, a_2, ..., a_n\}$. During training on downstream table reasoning datasets, we can choose to freeze the parameters of the LLMs and only learn the tabular encoder and alignment layers. *This method allows us to capture structure representations in the tabular modality while integrating them with LLMs in a **cost-effective** and **scalable** manner.*

## 3    EXPERIMENTS

In this section, we will demonstrate the advantages of treating tables as an independent modality (TAMO). Section 3.1 introduces our novel benchmark, StructQA, designed to evaluate LLMs' understanding of table structures and their robustness. Sections 3.2 and 3.3 present the performance gains of our approach across mainstream datasets and fine-tuning methods. Section 3.4 explores the interpretability of our method through attention visualization. Section 3.5 demonstrates the scalability of our approach to other LLMs. Section 3.6 showcases the robust performance of our method under different fine-tuning techniques. Finally, Section 3.8 provides an in-depth analysis of alignment details.

### 3.1    STRUCTQA: TABLE STRUCTURE UNDERSTANDING TASK

In this work, we propose to emphasize the importance of table structure in table reasoning and first establish an open-source evaluation benchmark *StructQA*, which consists of 5 types of table structure understanding tasks (Table 1) and 7500 question-answer pairs from 500 tables. More construction details can be found in Section B. Unlike conventional datasets, *StructQA* evaluates a model's structure understanding comprehensively across three dimensions: (i)-**direct performance**; (ii)-**permutation**: performance after randomly shuffling the rows and columns of tables in the test set; (iii)-**robustness**: consistency of answers before and after permuting, regardless of accuracy. Besides, the newly-released benchmark mitigates potential risks of data contamination (Ye et al., 2024b) present in existing publicly available datasets to a certain extent.

(1) ***Cell location***: identify cell value by row number and column name.

(2) ***Column lookup***: identify the column based on row number and cell value.

(3) ***Row lookup***: identify the row based on the column name and cell value.

(4) ***Column comprehension***: summarize all distinct values in a column based on the column name.

(5) ***Row comprehension***: summarize all distinct values in a row based on the row number.

Table 1: Five different types of structural tasks in the *StructQA* dataset. More details are in Appendix B.

### 3.2    EXPERIMENTAL SETUP

**Datasets & Metrics.** To evaluate the effectiveness of TAMO, we conducted extensive experiments on *StructQA* and four public table reasoning benchmarks. To examine the unique contributions of table embeddings for different tasks, we trained each TAMO separately on the training set of each respective task and evaluated it on corresponding test sets.

(i) ***HiTab*** (Cheng et al., 2022) features hierarchical tables with multi-level headers, comprising 10,672 questions over 3,597 tables. We use execution accuracy as the evaluation metric, demonstrating the superiority of hypergraphs in modeling hierarchical tables.

(ii) ***WikiTableQuestions*** (WikiTQ) (Pasupat & Liang, 2015) involves complex questions answering over 2,108 Wikipedia tables with 22,033 questions requiring complex reasoning and aggregation. The primary evaluation metric is answer accuracy compared to the ground truth.

(iii) ***WikiSQL*** (Zhong et al., 2017) focuses on natural language to SQL query generation, containing 80,654 questions paired with SQL queries over 24,241 Wikipedia tables. Execution accuracy measures the correctness of query results.

(iv) ***FeTaQA*** (Nan et al., 2022) emphasizes free-form question answering with comprehensive, free-text answers, featuring 10,279 questions over 3,641 Wikipedia tables. The BLEU metric is recommended officially to evaluate the similarity between generated and reference answers.

**Competing Methods.** To demonstrate that incorporating tabular modality into LLMs, referred to as *tabular language models*, can enhance performance in table reasoning tasks, we compare **TAMO** against using only pure text modality in four different settings: (i)-***Inference Only***: using LLMs to directly reason on serialized table sequences and questions. (ii)-***Frozen LLM***: comparing with prompt tuning (Lester et al., 2021), which adds some parameterized and trained tokens in front of serialized table sequences. (iii)-***Tuned LLM (LoRA)***: using LoRA (Hu et al., 2021) to finetune the parameters of LLMs. We add optional LoRA in our method as $\text{TAMO}^{+}_{LoRA}$. (iv)-***Tuned LLM (SFT)***:

| Setting | Dataset
Task Type
Evaluation Metric | StructQA
Structural QA
Accuracy | HiTab
Hierarchical QA
Accuracy | WikiTQ
Table QA
Accuracy | WikiSQL
Table QA
Accuracy | FetaQA
Free-form QA
BLEU |
|---|---|---|---|---|---|---|
| Inference Only | Zero-shot | 8.60 | 7.77 | 14.50 | 21.44 | 20.08 |
| Frozen LLM | Prompt tuning
**TAMO**
$\triangle_{Prompt\ tuning}$ | 37.80
59.07
↑ 56.27% | 26.26
48.86
↑ 86.06% | 29.86
37.06
↑ 24.11% | 61.24
76.45
↑ 24.84% | 29.94
36.52
↑ 21.98% |
| Tuned LLM
(LoRA) | LoRA
**TAMO**$^+_{LoRA}$
$\triangle_{LoRA}$ | 45.67
70.80
↑ 55.03% | 50.76
59.22
↑ 16.67% | 37.13
43.53
↑ 17.24% | 57.10
84.43
↑ 47.86% | 35.80
37.43
↑ 4.55% |
| Tuned LLM
(SFT) | TableLlama(2023b)
SFT
**TAMO**$^+_{SFT}$
$\triangle_{SFT}$ | 6.47
62.73
**71.60**
↑ 14.14% | 63.76
54.80
**63.89**
↑ 16.59% | 31.22
43.28
**45.81**
↑ 5.85% | 46.26
79.86
**85.90**
↑ 7.56% | 38.12
37.37
**39.01**
↑ 4.39% |
| Others | GPT-3.5
GPT-4
Specialist SOTA | 41.93
51.40
- | 43.62*
48.40*
64.71(2023b) | 53.13*
68.40*
69.10(2024) | 41.91*
47.60*
92.07(2022) | 26.49*
21.70*
40.50(2024) |

Table 2: Results on our table structure understanding dataset *StructQA* and four table reasoning benchmarks. TAMO adds additional table modality information compared to the pure text baseline. Specialist SOTA refers to methods that design models and training tasks specifically for each dataset. "*" indicates data sourced from Zhang et al. (2023b). The first best result for each task is highlighted in **bold** and the second best result is highlighted with an underline.

supervised finetuning of all parameters of LLMs. TAMO$^+_{SFT}$ means supervised training of TAMO and LLMs jointly.

Additionally, to comprehensively evaluate the ability of TAMO, we also compare with the *dataset-specific* state-of-the-art (SOTA) methods and evaluate the powerful black-box LLMs GPT-3.5-turbo-0125 & GPT-4-turbo-2024-04-09. TableLlama (Zhang et al., 2023b), derived from Llama2-7B through specialized fine-tuning on extensive tabular datasets, achieves SOTA performance on multiple tasks and is evaluated under the "Tuned LLM (SFT)" setting.

## 3.3 MAIN RESULTS

We evaluate the effectiveness of TAMO on our constructed table structure understanding dataset *StructQA* and four table reasoning benchmark datasets: HiTab, WikiTQ, WikiSQL, and FetaQA. The results are shown in Table 2. We consistently use Llama2-7B as the base LLM for our method and all baselines. Note that GPT-3.5, GPT-4, and specialist SOTA models are included only for reference and not for fair comparison.

**Explicitly inputting the tabular modality significantly enhances LLM's performance in various table reasoning tasks.** Across *all* datasets, whether it is table structure understanding task (StructQA), hierarchical table QA (HiTab), complex table QA (WikiTQ, WikiSQL), or free-form table QA (FetaQA), TAMO achieves substantial improvements in *both* frozen and tuned LLM settings. For example, TAMO shows an average improvement of **+42.65%** over inputting pure text modality on the frozen LLM setting, with a maximum improvement of **+86.06%** on the HiTab dataset. In the tuned LLM setting, both TAMO$^+_{LoRA}$ and TAMO$^+_{SFT}$ show substantial improvements, outperforming the pure text modality by an average of +28.27% and +9.71%, respectively.

Meanwhile, TAMO$^+_{SFT}$ achieves SOTA performance across all tasks under our settings. TAMO$^+_{LoRA}$ secures a close second on 3 out of 5 datasets and **significantly outperforms the SFT models that rely solely on the text modality**. This reveals the limited informational capacity of the pure text modality in table reasoning, highlighting that the table modality can provide a more comprehensive understanding. *Finally, all the above experimental results validate the feasibility of further enhancing the table comprehension and reasoning abilities of tabular LLMs by inputting global table structure information in a multimodal manner.*

**TAMO$^+_{SFT}$ is competitive with specialist SOTA methods, highlighting the utility of using hypergraphs to model complex table structure relationships.** The Llama2-7B based TAMO$^+_{SFT}$ achieves closed SOTA performance on HiTab, FetaQA, and WikiSQL, where HiTab is a complex

**Inference only**          **TAMO (Ours)**

Figure 5: A real visualization case in the WikiSQL dataset results of attention weights from other input tokens to the label answer cell "Canada". Intuitively, the darker the color, the more closely the token is associated with "Canada". We observe that with the "[table_structure_token]" of TAMO, the LLM better focuses on information relevant to the correct answer, as indicated by the darker background colors associated with those tokens.

hierarchical table dataset. This indicates that hypergraph-enhanced tabular encoder can effectively learn complex hierarchical relationships within tables, thus further improving the model's accuracy in table reasoning tasks. Although slightly behind the specialist SOTA methods on the other datasets, it's worth noting that they all utilized *dataset-specific* model architectures, training methods, or other enhancement tricks. In contrast, our approach is the first attempt to input tables as an independent modality into LLMs and delivers impressive *generalization* across various table reasoning tasks. Additionally, $\text{TAMO}_{LoRA}^{+}$ and $\text{TAMO}_{SFT}^{+}$ consistently surpass GPT-3.5 and GPT-4 on 4 out of 5 datasets. For example, it achieves an average improvement of over **+0.22** accuracy compared to GPT-3.5.

## 3.4 TAMO AS AN INTERPRETABLE LEARNER

To analyze the interpretable impact of the *table structure token* on LLMs' reasoning, we visualize the attention importance of all input tokens for the correct answer as perceived by the LLMs. Specifically, we adopt the visualization method from the PromptBench (Zhu et al., 2023b), which uses the gradients of the input embeddings to estimate token importance. We randomly select a sample from the WikiSQL test sets for visualization analysis, where the base method (inference only) is incorrect but TAMO is correct. The result is shown in Figure 5. We find: (i)-TAMO thinks "Canada" (correct answer) and "US HL" (relevant contextual information) tokens are the more important for the final answer, while the base method largely ignores these crucial tokens. (ii)-TAMO shows a certain level of attention to "[table_structure_token]", and adding "[table_structure_token]" affects the importance distribution of other input tokens, prompting LLMs to focus more on tokens relevant to the correct answer. We observed some error cases with the LoRA setting that resemble those shown above. For example, when the correct answer is far from the question in the serialized input, TAMO can utilize the overall table structure to locate the correct answer, compared to LoRA in text-only mode, which primarily focuses on the content immediately before and after the question. This case study indicates that ***the structural information in* TAMO *can improve the reasoning abilities of LLMs for tabular tasks.***

## 3.5 TAMO AS A SCALABLE LEARNER

To validate the scalability of the proposed TAMO across different LLMs, we experimented with TableLlama (Zhang et al., 2023b) and Mistral-7B on the frozen LLM setting, in addition to Llama2. The experimental results, as shown in Table 3, demonstrate significant improvements for *both* TableLlama and Mistral-7B with TAMO compared to the pure text modality. Specifically, TAMO improves performance by **26.99%** on TableLlama. These results confirm TAMO's scalability across different LLMs.

Additionally, we observed the following findings in Table 3: (i)-The minimal gap (0.0016 acc.) between

| Method | Llama2 | TableLlama | Mistral |
|---|---|---|---|
| Inference Only (Base) | 14.50 | 31.22 | 18.44 |
| Prompt tuning | 29.86 | 31.38 | 44.98 |
| **TAMO** | **37.06** | **39.85** | **47.33** |
| $\triangle_{Prompt\ tuning}$ | ↑ 24.11% | ↑ 26.99% | ↑ 5.22% |

Table 3: Evaluate the scalability for different LLMs of our proposed TAMO on the frozen LLM setting (prompt tuning) on the WikiTQ dataset.

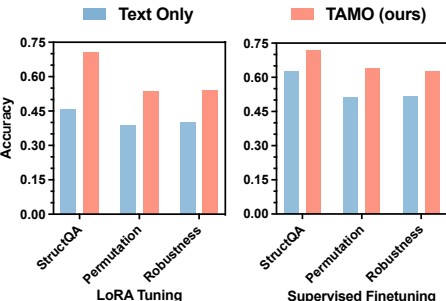 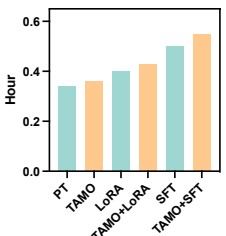 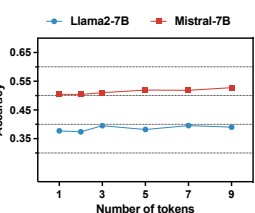

Figure 6: Evaluate the robustness of TAMO to permutation invariance on the StructQA dataset. *Permutation*: randomly permuting rows and columns in the StructQA test set. *Robustness*: the proportion of samples that remain consistent after random permutation.

Figure 7: Training time efficiency comparison under different settings for 1 epoch on WikiTQ dataset.

Figure 8: Analysis study of different numbers of table structure tokens on the WikiTQ dataset.

the base and prompt tuning on TableLlama indicates that the supervised fine-tuned LLMs already possess a strong capability to follow tabular format instructions. Consequently, prompt tuning has a limited effect. However, **incorporating global tabular structure information through TAMO further enhances table reasoning capabilities.** (ii)-The ultimate performance of TAMO is influenced by the capability of the LLMs. For instance, Llama3 shows significantly better performance than TableLlama (based on Llama2).

### 3.6 TAMO AS A ROBUST LEARNER

Compared to image/text data, *permutation invariance*—any permutation of the rows and columns does not change the original interpretation of the table—is a unique structural property of tabular data. To further explore whether TAMO can effectively perceive table structure information, we construct experiments to assess its robustness regarding permutation invariance. Specifically, we use the permutation version test set by randomly shuffling the rows and columns of tables in the StructQA test set (the training set is unchanged). In the frozen LLM setting, we compare the performance of TAMO with pure text modality methods (inference only & prompt tuning) on the new test set and check the consistency of answers after permutation. Results are shown in Figure 2 and Figure 6, we find that for *both* frozen LLMs and tuned LLMs (LoRA and SFT), TAMO consistently outperforms pure text modality methods. Additionally, TAMO demonstrates the best robustness in maintaining consistent results after permutation. These indicate that TAMO effectively inputs table structure information into LLMs through our proposed multimodal method, enhancing their performance on tabular tasks.

### 3.7 TAMO AS AN EFFICIENT LEARNER

To further demonstrate the practicality of TAMO, we evaluate its operational efficiency. In our experiments, we utilize a server equipped with 2 H100 GPUs. Only SFT uses 2 GPUs while conducting all other experimental setups with single GPU training. We measure the time required to run 1 epoch on the WikiTQ dataset. The results are shown in Figure 7. We found that (i)-TAMO has a faster runtime efficiency compared to LoRA; (ii)-TAMO$^+_{LoRA}$ shows only a slight increase in runtime compared to LoRA, as does TAMO$^+_{SFT}$ compared to SFT. Therefore, injecting learnable table features does not significantly add to the computational burden in practical applications.

### 3.8 ANALYSIS STUDY

We further explore the impact of the table structure token quantity parameter on the model's performance. Specifically, in the frozen LLM setting, we evaluate TAMO on the WikiTQ dataset with varying numbers of table structure tokens. Due to limited computational resources, we randomly selected 6000 samples from the WikiTQ training set for the experiments, keeping the validation

and test sets unchanged. The experimental results are shown in Figure 8. The final performance of the model is consistently similar when the number of tokens is two or more $\{2, 3, 5, 7, 9\}$, which indicates that a minimum of two tokens is sufficient to explain the structural information in the table.

# 4 RELATED WORK

**LLM-based Table Reasoning.** Recently, with the rapid development and outstanding performance of Large Language Models (LLMs), LLM-based methods have become the mainstream approach for tabular reasoning tasks (Zhang et al., 2024), collectively known as Tabular Large Language Models. These methods fall into two main categories: *(i) Fine-tuning on Tabular Data*: This approach enhances LLMs' understanding and reasoning abilities on structured data through supervised fine-tuning on tables (Zhang et al., 2023b; Zhuang et al., 2024; Wu & Feng, 2024; Sarkar & Lausen, 2023). For example, TableLlama (Zhang et al., 2023b) fine-tunes Llama2-7B on various real-world tables to create a generalist model for tables. *(ii) Prompt Engineering for Specific Table Tasks*: This approach uses specially designed prompts to enhance LLMs' reasoning capabilities in specific scenarios (Ni et al., 2023; Wang et al., 2024; Jiang et al., 2023b; Zhang et al., 2023b; Cheng et al., 2023). For instance, Dater (Ye et al., 2023) improves reasoning accuracy by decomposing large tables into smaller subtables with multi-step prompts, while Chain-of-table (Wang et al., 2024) uses chain-of-thought and programming language-like methods for complex tabular problems.

**Table Encoder.** In recent years, numerous studies have explored effective methods for encoding and understanding tabular data. Yin et al. (2020) adopts a dual-encoder framework that separately processes textual and structural elements of tables, improving table comprehension through masked language modeling. Chen et al. (2024) extends this concept by using hyperedges to capture richer interactions among simple flat table cells, resulting in enhanced representations for relational data. Arik & Pfister (2021) utilizes a novel iterative masking attention mechanism to select important features. However, all these table encoders cannot handle joint text and table understanding tasks like table question answering. They are primarily used to encode raw tabular data into a low-dimensional vector space to get better table representation. As discussed in Section 1, inputting tables into tabular LLMs is challenging, as traditional methods serialize tables into text sequences, losing global structure. We propose a novel multimodal approach to help LLMs understand both structural relationships and textual semantics, enhancing their reasoning capabilities for tabular tasks.

# 5 LIMITATIONS

While our framework, TAMO, enhances frozen-parameter LLMs' understanding of tabular data through hypergraph encoders and learnable features, it has certain limitations. First, it relies on pre-structured tables, as required by the TableQA paradigm (Pasupat & Liang, 2015). For tables embedded in unstructured text, text-to-table techniques (Wu et al., 2022; Deng et al., 2024) are needed to structure the data. Second, unlike large visual multimodal models (Liu et al., 2023; Zhu et al., 2023a) that leverage pre-trained visual-text encoders like CLIP (Radford et al., 2021), there is currently no large-scale pre-trained table modality encoder aligned with LLMs. Our work provides a preliminary demonstration that table modalities can be independently encoded and understood by LLMs. Finally, extensive modal instruction data is required to develop robust, out-of-the-box multimodal capabilities, which we leave for future work. These limitations highlight the early stage of our research and the need for further exploration to fully integrate table modalities with LLMs.

# 6 CONCLUSION

In this work, we introduced a novel framework, TAMO, which leverages a hypergraph-enhanced tabular encoder to boost frozen-parameter LLMs' understanding of tabular data. By adhering to the principle of table structure permutation invariance, TAMO effectively encodes table structures into LLM-comprehensible representations using learnable features. This enables the handling of tasks involving both text and table understanding, such as table QA. Additionally, we presented StructQA, a dataset focused on table structure understanding, and validated our framework's efficacy and versatility across four other public table QA benchmarks.

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

## A  ETHICS STATEMENT

Our research endeavors to advance the capabilities of Large Language Models (LLMs) in understanding and processing tabular data, aiming for broader applicability and enhanced accuracy via simulated human-like table reasoning. We are committed to conducting this research ethically and responsibly. The datasets used in our experiments are publicly available and sourced in a manner that respects data privacy and intellectual property rights. We acknowledge the potential societal impacts of advanced AI systems and strive to ensure that our work promotes positive outcomes.

However, we recognize the risks associated with the misuse of powerful AI technologies, including privacy violations, biased decision-making, and the potential for reinforcing existing inequalities. To mitigate these risks, we advocate for transparency, fairness, and accountability in the development and deployment of AI systems. We also encourage continuous dialogue with the broader community to address ethical concerns and foster the responsible use of AI advancements.

By emphasizing these principles, we aim to contribute positively to the field of AI while remaining vigilant about the ethical implications of our work.

## B  STRUCTQA DATASET DETAILS

As mentioned in Section 3.1, we construct a table structure understanding dataset **StructQA**, which has 5 types of table structure tasks. Here, we provide the construct details. Specifically, we randomly select 500 tables from WikiTQ (Pasupat & Liang, 2015), creating 3 question templates for each table per task, resulting in 7500 question-answer pairs. We split the data into training, validation, and test sets with a ratio of 60%, 20%, and 20%, respectively. The question templates for each task are as follows:

(1) ***Cell location***

- What is the value in the column {column name} of sample row {row number}?

- Can you tell me the value of the column {column name} in sample row {row number}?

- In sample row {row number}, what is the value for the column {column name}?

(2) ***Column lookup***

- In sample row {row number}, which columns contain the value {cell value}?

- Can you identify the columns in sample row {row number} that have the value {cell value}?

- Which columns in sample row {row number} are associated with the value {cell value}?

(3) ***Row lookup***

- Which rows in the column {column name} have a value of {cell value}?

- Can you identify the sample rows where the column {column name} equals {cell value}?

- In the column {column name}, which rows contain the value {cell value}?

(4) ***Column comprehension***

- What are the distinct values in the column {column name}?

- Could you list the unique values present in the column {column name}?

- In the column {column name}, what various values can be found?

(5) ***Row comprehension***

- What are the values of each cell in row {row number} of the sample?

- Could you provide the cell values for each column in sample row {row number}?

- In sample row {row number}, what are the respective cell values?

## C EXPERIMENTS

### C.1 IMPLEMENTATION SETTINGS

Experiments are conducted using 2 NVIDIA H100-80G GPUs. Each experiment is replicated four times, utilizing different seeds for each run to ensure robustness and reproducibility.

**LLM.** We use the open-sourced Llama2-7b[2] as the LLM backbone. In fine-tuning the LLM with LoRA, the lora_r parameter (dimension for LoRA update matrices) is set to 8, and the lora_alpha (scaling factor) is set to 16. The dropout rate is set to 0.05. In prompt tuning, the LLM is configured with 8 virtual tokens. The number of max text length is 1024. The number of max new tokens, the maximum number of tokens to generate, is 128. We use Mistral-7B[3] for some experiments.

**Optimization.** We use the AdamW optimizer. We set the initial learning rate at 1e-5, with a weight decay of 0.05. The learning rate decays with a half-cycle cosine decay after the warm-up period. The batch size is 8, and the number of epochs is 10. To prevent overfitting and ensure training efficiency, an early stopping mechanism is implemented with a patience setting of 3 epochs.

### C.2 EVALUATION OF LEARNED HYPERGRAPH REPRESENTATION

To evaluate the effectiveness of the learned hypergraph representations, we conducted additional experiments by adding an MLP classifier head to predict table structure. Specifically, we used a binary classification task to predict whether a given cell in the table belongs to a specific row or column. The dataset for this task was derived from the WikiTQ (Pasupat & Liang, 2015) dataset, using its training, validation, and test table splits to construct corresponding samples. And the metric is the F1 score. The experiments, all trained for 50 epochs with a learning rate of 3e-4, produced the following results shown in Table 4:

| Settings | F1 Score |
|---|---|
| MLP head | 5.39 |
| + randomly initialized hypergraph | 49.73 |
| + pretrained hypergraph of TAMO | |
| StructQA | **71.32** |
| HiTab | 66.39 |
| WikiTQ | 62.63 |
| WikiSQL | 68.00 |
| FetaQA | 64.99 |

Table 4: Evaluation of the hypergraph representation to predict table structure.

- **MLP Classifier Without Hypergraph Representation:** To establish a baseline, we evaluated a model with only an MLP classifier without any hypergraph input. This setup performed poorly, achieving an F1 score of merely **5.39%**, underscoring the necessity of hypergraph representations for capturing table structure.

- **Random Initialization of the Hypergraph Network + MLP Classifier:** In this setup, we trained a classifier on a randomly initialized hypergraph network combined with an MLP head to assess whether the structure could be learned from scratch. This approach achieved an F1 score of **49.73%**, indicating some ability to learn structure but highlighting the challenges without prior knowledge.

- **Pretrained Hypergraph Network of TAMO from each dataset + MLP Classifier:** In this experiment, we used the hypergraph network pretrained on each dataset (i.e., StructQA, HiTab, WikiTQ, WikiSQL, and FetaQA) with an MLP classifier. All models achieved F1 scores above 60%, with StructQA achieving the highest score of **71.32%**, likely due to its lower reasoning

---

[2]https://huggingface.co/meta-llama/Llama-2-7b-hf
[3]https://huggingface.co/mistralai/Mistral-7B-v0.1

complexity, which allows for more focused table structure representations by minimizing irrelevant noise. These results demonstrate that TAMO's hypergraph embeddings effectively encode structural relationships and generalize across datasets, as all evaluations were conducted on the WikiTQ test set, distinct from the pretraining datasets. And they can recover table structure with high accuracy.

Based on these experiments and the interpretability analysis in Section 3.4, we believe hypergraph-based representations help LLMs understand table structures and locate answers more effectively during reasoning—a critical capability for TableQA, as also validated in previous work (Yang et al., 2022).

### C.3 EVALUATION OF CROSS-DATASET GENERALIZATION IN TAMO

In Table 2, we demonstrated that TAMO, when trained individually on each dataset, achieves significant improvements on the corresponding test sets. This raised the question of whether TAMO's table structure embeddings are generalizable to other datasets. To address this, we evaluated TAMO models trained on one dataset against the test sets of other datasets, as shown in Table 5.

Theoretically, TAMO's table structure embeddings are designed to model general table structures. However, the training process also relies on task-specific instruction data, and the loss for learning table structure representations is tied to QA objectives. **This means the embeddings can be influenced by the types of instructions used during training, introducing task-specific biases.** For example, embeddings trained on StructQA, which involves simpler table structures, tend to perform well on structural recognition tasks but lack the complexity required for reasoning-heavy tasks like WikiTQ. Consequently, while table structure embeddings trained on individual tasks consistently outperform baselines without structure embeddings, they fall short of matching the performance of embeddings trained directly on the target task. We also observed that datasets with significant differences, such as FetaQA—which uses BLEU as an evaluation metric for free-text answers—show limited cross-dataset transferability. The model trained on FetaQA fail to provide improvements on other datasets, and vice versa. However, for QA datasets with similar formats and objectives, such as WikiTQ and WikiSQL, we observed some degree of transferability, suggesting that TAMO can leverage shared patterns among related tasks. This observation is consistent with findings in TableLlama (Zhang et al., 2023b), where differences in task formats and reasoning complexity limited cross-task generalization.

| Evaluation Dataset | StructQA | HiTab | WikiTQ | WikiSQL | FetaQA |
| Metric | Accuracy | Accuracy | Accuracy | Accuracy | BLEU |
| --- | --- | --- | --- | --- | --- |
| Base | 8.6 | 7.77 | 14.5 | 21.44 | 20.08 |
| StructQA | **59.07** | 16.73 | 18.74 | 32.57 | 8.38 |
| HiTab | 17.53 | **48.86** | 27.46 | 38.83 | 1.78 |
| WikiTQ | 16.40 | 29.29 | **37.06** | 38.74 | 0.95 |
| WikiSQL | 18.73 | 24.43 | 23.85 | **76.45** | 1.18 |
| FetaQA | 0.00 | 0.00 | 0.02 | 0.00 | **36.52** |

Table 5: Generalization results of each TAMO separately trained on different dataset.

To isolate the effect of table structure representations from task-specific biases, we conducted additional experiments focusing solely on table structure prediction tasks. As shown in Table 4, table encoder trained on one dataset achieved F1 scores above **60%** on structure prediction tasks from the other dataset. This demonstrates that TAMO's table encoder captures a unified representation of table structures and validates the generalizability of our approach.

A key factor is the absence of large-scale, task-agnostic pretraining for TAMO's table encoder. Similar to how CLIP (Radford et al., 2021) decouples modality-specific representations through extensive pretraining, a dedicated pretraining phase for TAMO's table encoder—focusing purely on table-related structural information—could mitigate task-specific biases. This remains an important direction for future work to enhance generalization across domains and datasets.

## C.4 EFFECTIVENESS ON MULTIPLE-TABLE SCENARIOS

To validate TAMO in multiple-table scenarios, we have conducted additional experiments on the MultiTabQA-geoQuery (Pal et al., 2023) dataset. This dataset involves multiple-table queries with total token lengths reaching up to 4K, relatively larger than current TableQA benchmarks. Specifically, we evaluated its cell selection task using precision, recall, and F1 score as metrics. Due to the unique output format requirements of this task, we adopted a one-shot setting across the following experiments while keeping other parameters unchanged. As shown in Table 6, TAMO achieves over 40% and 100% improvements under frozen LLM and SFT LLM settings, respectively, demonstrating its effectiveness in multi-table scenarios. While TAMO shows only marginal advantages in the LoRA setting, we will investigate the detailed configurations in future work.

| Setting | Method | Precision | Recall | F1 score |
|---|---|---|---|---|
| Inference Only | One-shot | 9.68 | 5.96 | 7.38 |
| Frozen LLM | Prompt tuning | 4.83 | 3.46 | 4.03 |
| | **TAMO** | 6.82 | 4.86 | 5.67 |
| | $\triangle_{Prompt\ tuning}$ | ↑41.20% | ↑40.46% | ↑40.69% |
| Tuned LLM (LoRA) | LoRA | 30.56 | 10.30 | 15.41 |
| | **TAMO**$^+_{LoRA}$ | 28.32 | 10.67 | 15.50 |
| | $\triangle_{LoRA}$ | ↑−7.33% | ↑3.59% | ↑0.58% |
| Tuned LLM (SFT) | SFT | 30.55 | 11.04 | 16.22 |
| | **TAMO**$^+_{SFT}$ | **49.36** | **25.46** | **33.59** |
| | $\triangle_{SFT}$ | ↑61.57% | ↑130.62% | ↑107.09% |

Table 6: Effectiveness on MultiTabQA-geoQuery.

## C.5 CHOICE OF BACKBONE MODEL

Our motivation stemmed from observing the limited robustness of structure recognition in TableLlama (Zhang et al., 2023b), a LLaMA2-based model, in table-related tasks. For consistency in experimental settings, we also chose LLaMA2 7B as our backbone and successfully demonstrated that even with the relatively lower-performing LLaMA2, the addition of our hypergraph encoder led to substantial performance improvements.

We further validate TAMO on more advanced open-source LLMs. Due to computational constraints, we conducted frozen-LLM experiments with LLaMA 3.1 8B, as shown in Table 7. The results indicate that while LLaMA 3.1 8B achieves a stronger baseline than LLaMA 2 7B, adding the table encoder consistently improved performance, with gains reaching over 10% on certain datasets. This further validates the unique benefits of hypergraph-based structural representation of tables across more advanced open-source LLMs.

| Setting | Dataset
Task Type
Evaluation Metric | StructQA
Structural QA
Accuracy | HiTab
Hierarchical QA
Accuracy | WikiTQ
Table QA
Accuracy | WikiSQL
Table QA
Accuracy | FetaQA
Free-form QA
BLEU |
|---|---|---|---|---|---|---|
| Inference Only | Llama 3.1 8B | 15.73 | 19.51 | 23.80 | 31.60 | 14.05 |
| Frozen LLM | Prompt tuning | 71.53 | 69.38 | 53.71 | 77.06 | 36.16 |
| | **TAMO** | 78.00 | 73.73 | 56.93 | 85.44 | 38.09 |
| | $\triangle_{Prompt\ tuning}$ | ↑9.05% | ↑6.27% | ↑6.00% | ↑10.87% | ↑5.34% |

Table 7: Results on advanced LLM.

# D DISCUSSIONS

## D.1 POSITIONING OF TAMO

While both HyTrel (Chen et al., 2024) and TAMO adopt a hypergraph-based framework, there are significant distinctions. HyTrel focuses on general tabular representation learning and, as stated in its limitations, cannot handle joint text-table reasoning tasks like TableQA. In contrast, it is non-trivial for TAMO to pioneer treating tables as an independent modality within LLMs, aligning hypergraph-based table representations with text representations to tackle complex reasoning tasks.

This distinction parallels advancements in other domains. For example, in vision, ViT (Dosovitskiy et al., 2020) and CLIP Radford et al. (2021) act as modality encoders, while GPT-4v (OpenAI, 2023) and LLaVA (Liu et al., 2023) integrate these encodings into multimodal frameworks. In the audio domain, there is a similar phenomenon, as shown in Table 8. For the first time, TAMO fills this gap in the table domain, going beyond a table encoder to a multimodal reasoning framework. This cross-modal fusion makes TAMO a significant advancement, not an incremental improvement. Notably, while TAMO and HyTrel share a similar network architecture, their training tasks and optimization objectives are entirely different, further underscoring the contribution of our approach.

| Domain | Modality Encoder | Multimodal LLMs |
|---|---|---|
| Vision Domain | ViT (2020), CLIP (2021) | GPT-4v (2023), LLaVA (2023), MiniGPT-4 (2023a) |
| Audio Domain | Whisper (2023) | SpeechGPT (2023a), AudioPaLM (2023) |
| Table Domain | HyTrel (2024) | **TAMO (Ours)** |
| Role | Encoding domain-specific data | Modality alignment with LLMs to obtain corresponding domain-specific multimodal models |
| Ability for Generative Tasks (e.g., QA) | No | **Yes** |

Table 8: Positioning of TAMO in the table domain. TAMO is the first multimodal LLM designed for the table domain.

## D.2 COMPARISON WITH POTENTIAL APPROACHES

We acknowledge that there are several alternative methods to model table structures effectively, such as using 2D positional embeddings to capture row and column information and data augmentation techniques to enforce permutation invariance. Below, we discuss these methods in the context of their applicability and limitations accordingly.

**Using 2D positional embeddings to capture row and column information.** Using 2D positional embeddings is indeed a natural approach, as it captures row and column information directly. However, implementing this method often requires intrusive modifications to the position encoding layer of LLMs (e.g., as in TableFormer (Yang et al., 2022)), demanding extensive re-training of these position encodings. Such re-training is highly dependent on specific LLM architectures, and the learned modifications are not theoretically transferable to other LLMs. In contrast, our proposed table encoder is designed to **operate as an external plugin of tabular modality, minimizing modifications to the LLM itself.**

**Data augmentation techniques to enforce permutation invariance.** While data augmentation techniques to enforce permutation invariance are intuitive, they present practical challenges. For tables with dimensions $n \times m$, the number of possible permutations grows factorially as $n! \times m!$. Training on such a large augmented dataset is computationally prohibitive, and the resulting models are prone to overfitting due to the enormous training data requirements. TAMO is designed to **be**

**data-efficient, achieving structural permutation invariance without relying on large-scale data augmentation.**

As illustrated in Appendix C.1, the objective of our work is to establish the feasibility of treating structured data as a distinct modality modeled through a dedicated table encoder. By doing so, we enable a modular and flexible integration of tabular data across diverse architectures. While potential methods, such as 2D positional embeddings and data augmentation, are valuable, they are outside the scope of this study and represent potential directions for future work.

### D.3 BEYOND ROW AND COLUMN PERMUTATIONS

While row and column permutations are the most prominent cases in tabular data, other forms of order permutations can arise in more complex table structures. These include:

**Nested Table Structures.** In hierarchical or grouped tables, sub-tables are often nested within a broader table structure. Permutations can occur within these nested sub-tables, reflecting changes in the ordering of hierarchical levels. Such structures are common in multi-level reports and datasets with grouped summaries.

**Composite Attributes.** Tables may contain multi-column attributes where relationships or dependencies exist between columns. For instance, in a table representing geographic data, attributes such as latitude and longitude might form a composite structure. Permutations within such attributes could represent alternative orderings of these dependent fields, requiring specialized handling to maintain semantic coherence.

**Cell-Level Permutations.** In some cases, individual cells may contain structured or semi-structured data, such as lists, arrays, or key-value pairs. Order changes within these cell values represent another form of permutation, particularly relevant in domains where embedded structured data is prevalent (e.g., JSON-like entries or lists of items within a cell).

While these forms of permutations are significant in certain contexts, they are most commonly observed in complex hierarchical datasets, such as HiTab (Cheng et al., 2022). In this study, we focus primarily on flat table structures from mainstream TableQA datasets, where row and column permutations are the predominant concerns. Addressing these additional forms of permutation is an important direction for future work, particularly for datasets with more complex organizational patterns.

