# OpenReview forum: "Injecting Learnable Table Features into LLMs"
_ICLR.cc/2025/Conference — Submitted to ICLR 2025_

### Official Review · Reviewer_Vyb4 · 2024-10-21

**Soundness:** 2
**Presentation:** 3
**Contribution:** 2
**Rating:** 5
**Confidence:** 4

**Summary:**

This paper proposed TAMO (Table as Modality), to inject tabular representations as a separate modality into LLMs to enhance model's robustness towards order permutation and performance on table reasoning tasks. Current table LLMs typically rely on serializing table data while ignoring the relationships between rows and columns and the overall table structure. TAMO uses a hypergraph-enhanced tabular encoder to aggregate structural information by doing average-pooling over cells thus making it robust to order permutations. This structured feature is further injected as a prefix soft prompt, followed by the serialized textual input of the table and the question. The authors also propose StructQA, a new benchmark designed to evaluate table structure understanding and robustness to structural perturbations, such as random row or column shuffling. Results on StructQA and other table QA datasets (HiTab, WikiTQ, etc) demonstrate that TAMO significantly improves performance over existing baselines, achieving a 42.65% average gain. Ablations on different aspects (interpretability, scalability, etc) also provides details for better understanding the proposed approach.

**Strengths:**

S1. The paper introduces a novel framework, TAMO, that treats tables as an independent modality, effectively addressing the limitations of existing LLMs in handling table reasoning tasks.

S2. The paper proposed StructQA, a benchmark for table structure understanding and robustness, which adds value and contributes to future research.

S3. TAMO shows superior performance on both StructQA and previous TableQA benchmarks, indicating the effectiveness of the proposed method.

S4. The paper is well-written and easy to follow.

**Weaknesses:**

W1. Method novelty: The hypergraph idea for tabular representation has been used in "HyTrel: Hypergraph-enhanced Tabular Data Representation Learning". Although HyTrel focuses on pretraining a hypergraph-based tabular representation for downstream tasks while TAMO focuses on TableQA/reasoning tasks, the idea seems to be not that novel and the work seems incremental.

W2. Computational complexity: TAMO used an additional module to encode the hypergraph. This requires to encode every cell/row/column in a table. Given that tables often contain millions of rows and thousands of columns, this can lead to significant computational and memory overhead. This paper appears to lack a detailed complexity analysis regarding the overall running time and memory usage of the method.

W3. Limited scenarios: Although the proposed method can indeed better capture the structural tabular information, the context length used in the experiments are relatively short (1024), which limits its practical usage. Also, since all experiments are done on single and short tables, I wonder whether or not this approach can perform well on huge tables (with millions of tokens) or on multiple-tables.

W4. Since this method requires to append embeddings in front of the input, it can only be used with open-sourced models but cannot be used on closed-sourced LMs like GPT-4.

**Questions:**

Q1. Are there any other cases for order permutations other than row/column permutations? I think it will increase the value of this work if more practical cases of order permutations can be named.

Typos:
Line 509~510: "...all these table encoders cannot cannot handle"

---

### Official Review · Reviewer_BCkd · 2024-10-29

**Soundness:** 2
**Presentation:** 1
**Contribution:** 3
**Rating:** 5
**Confidence:** 3

**Summary:**

The paper introduces TAMO (Treating Tables as an Independent Modality), a novel framework that treats tabular data as a separate modality integrated with Large Language Models (LLMs). The method embeds a table through the tabular encoder, that imposes a hypergraph structure on cells, and sets of semantically connected cells (rows or columns). The resulting embeddings are then used as soft prompts and can be injected into a wide array of publicly available LLMs. The paper also studies the robustness of other methods on a custom-made dataset.

**Strengths:**

## Originality
This approach is innovative, moving beyond the traditional method of serializing tables into text sequences. By using hypergraph neural networks to encode table structures, the paper offers a fresh perspective on table reasoning tasks, an understudied subject in the context of LLMs.

## Quality
The authors conduct experiments on multiple benchmark datasets, probably on all the meaningful ones in that area, including HiTab, WikiTQ, WikiSQL, FeTaQA, and the newly proposed StructQA. The empirical results demonstrate strong improvements over existing methods, indicating the effectiveness of TAMO. The inclusion of the StructQA dataset also contributes to the field by providing a benchmark focused on table structure understanding.

## Significance (Positive Aspects)
Addressing the limitations of LLMs in handling tabular data is a significant contribution. By treating tables as an independent modality, the paper opens up new possibilities for integrating structured data into language models. I think this makes the contribution much stronger.

## Clarity (Positive Aspects)
The paper provides a clear motivation for the problem, highlighting the limitations of current methods that serialize tables into text. Figures and diagrams are used to illustrate key concepts, such as the hypergraph representation of tables and the overall framework of TAMO.

**Weaknesses:**

## Repetition and Overstatement
The paper tends to repeat the same points, such as the limitations of serializing tables into text and the novelty of treating tables as an independent modality. Cutting down these sections would improve readability.
PLease reduce repetition by consolidating points about the limitations of existing methods and the benefits of the proposed approach. Also, make sure, you focus on presenting new information in each section to maintain the reader's engagement.


## Lack of Comprehensive Comparison:
The contributions are occasionally overstated without sufficient evidence. For instance, claiming that the hypergraph approach is superior without comprehensive comparisons to other methods.
The paper does not adequately compare the hypergraph-based method with other potential approaches that could model table structures, such as:
 - Using 2D positional embeddings to capture row and column information.
 - Data augmentation techniques to enforce permutation invariance.

Without these comparisons, it is difficult to conclude that the hypergraph approach is definitively better than other methods. It does in fact beat the naive baselines, but it does not prove to be any better than possible trainable alternatives with similar or lower complexities. It is also not clear from the paper how the TaMo is being trained. Is it trained once on all tasks or for each task separately?


## Methodological Complexity:
The introduction of the hypergraphs concept may add unnecessary complexity. If the primary goal is to encode rows and columns separately, and then iteratively improve the representations this could potentially be achieved with simpler narrative.
The explanation of the hypergraph construction and the multiset functions could be simplified. Providing intuitive explanations alongside mathematical formulations would aid understanding.
Please, explain why hypergraphs are used and how they benefit the model.


## Experimental Details:
The experimental setup lacks details in some areas:
Hyperparameter choices for the hypergraph encoder are not fully explained.
It is unclear how the model scales with larger tables or more complex structures.
Including ablation studies to show the impact of different components of the model would strengthen the experimental section.



# Writing

One significant weakness of the paper is the frequent misuse of terminology and unclear explanations, which negatively impact its clarity and hinder comprehension.

## Misuse of Terminology

For example, in **Section 2.3**, the authors state:

> "...we serialize tabular data into formatted text sequences and obtain the text tokens of tabular data \(X_{tt} \in \mathbb{R}^{L_s \times d_l}\) through the LLMs’ embedding layer,"

where they mistakenly use *"tokens"* instead of *"embeddings"* or *"representations."* This confusion between discrete tokens and continuous vector embeddings can mislead readers about the nature of the data being processed.

## Unclear Explanations

Similarly, ambiguous statements like:

> "This means that currently, when LLMs are involved in training, the model tends to focus more on the textual modality input"

lack context and clarity, making it difficult to understand the authors' intent.

## Grammatical Issues

Overcomplicated sentences and misuse of prepositions also hinder understanding. Also, there are multiple of sentence that are broken:
> "TAMO+_SF T is competitive with specialist SOTA methods and the superiority of using hypergraphs to model complex table structure relationships."
or
> "This approach incorporates both high-order hierarchical structure and permutation invariance as inductive biases, can perfectly model the complex structural properties in tabular data."

## Impact on Readability
These issues are not merely grammatical but relate to the precise use of technical language, which is essential for conveying complex ideas effectively. The lack of clear transitions between ideas and incomplete explanations of figures and tables disrupt the narrative flow, making it challenging to follow the authors' arguments.
Overall, these language and clarity issues significantly detract from the paper's quality and obscure its contributions. I think a careful revision to enhance readability and comprehension is necessary.

# Significance (Negative Aspects)
In many real-world applications, tables are embedded within unstructured text, such as in Wikipedia articles or research papers, where they are not readily available as separate, structured inputs. This reliance on pre-structured tables limits the applicability of TAMO, as it assumes that the table data is already parsed and organized in a way that allows for explicit definition of hyperedges in the hypergraph model. The necessity for structured table inputs means that the method may not be practical in scenarios where tables need to be extracted from raw text or where the table structure is ambiguous or inconsistent.

**Questions:**

1. How is the Tamo trained: is it a generic model, or is it just fine-tuned in a supervised manner for each task? Is there any custom change that needs to be specified for each task separately, or one trained model can handle all the datasets.
2. Is there a way to apply it beyond the table reasoning task?

---

### Official Review · Reviewer_aQGU · 2024-11-03

**Soundness:** 3
**Presentation:** 3
**Contribution:** 3
**Rating:** 5
**Confidence:** 4

**Summary:**

1. This paper introduces a new dataset, StructQA, which demonstrates that most existing LLMs exhibit poor robustness when textualized tables are used as input.

2. The paper proposes treating tables as a distinct modality, modeling tables using Hypergraph, and introducing TaMo,  which employs an encoder to encode Hypergraph aligning with LLMs.

3. Across four mainstream datasets, TaMo achieves significant performance improvements. Further analytical experiments demonstrate that TaMo exhibits stronger robustness compared to using textualized inputs.

**Strengths:**

1. The author addresses a critical issue, as the current use of textual tables as input for LLMs indeed affects their performance in table reasoning tasks.

2. The dataset proposed by the author, StructQA, can effectively assist in evaluating the table comprehension abilities of existing LLMs.

3. The author’s approach, TaMo, significantly enhances table reasoning performance.

**Weaknesses:**

1. Previous work has also proposed encoding tables with graphs to facilitate understanding of table structure [1][2]. Although these methods are based on small-scale models, directly replacing small-scale models with LLMs appears somewhat incremental.

2. The experiments are insufficient, resulting in unreliable performance for the proposed method. For further details, see the Questions section.

[1] https://arxiv.org/pdf/2309.11049

[2] https://arxiv.org/pdf/2209.07692

**Questions:**

1. In Section 2.3, authors utilize both the Hypergraph and Textual tables as inputs. Why is it necessary to input the textual table additionally? The author should discuss this choice and provide experimental evidence to support their viewpoint.

2. Compared to previous methods that directly model tables as graphs, what advantages does the author's proposed Hypergraph approach offer?

3. In Table 2, what training data are used for the baseline and TaMo? Is it StructQA, the training sets of the four evaluation datasets, or a combination of both?

4. In Section C.1, the author indicates that Llama2-7b is used as the backbone. Why not use more advanced open-source LLMs, such as Llama3.1-8b, which has better performance and might possess improved table reasoning capabilities?

***If the author could address these questions and incorporate responses into the paper, I would be glad to increase my rate.***

---

### Official Review · Reviewer_WgPY · 2024-11-05

**Soundness:** 3
**Presentation:** 4
**Contribution:** 3
**Rating:** 8
**Confidence:** 5

**Summary:**

The paper proposes TAMO, which treats table representation as an independent modality to enhance large language models (LLMs) reasoning capabilities over tables. TAMO leverages a theoretically permutation-invariant hypergraph to capture the table’s structure. In this hypergraph, nodes represent table cells, and hyperedges represent table headers, capturing the hierarchical structure of tables. By iteratively updating the representation of the nodes and hyperedges at each layer, TAMO learns the table’s structure and integrates this representation into the LLM’s hidden layers.

The authors also introduce a synthetic dataset, StructQA, to evaluate LLM robustness against structural permutations within tables. Experimental results show that TAMO training improves LLM performance on the TableQA task compared to supervised finetuning and demonstrates effectiveness on StructQA.

**Strengths:**

1. The paper introduces TAMO, a novel approach to learning table structures using hypergraphs, which supports both simple and complex table schemas.
2. The authors provide comprehensive experiments on four table QA datasets and the newly proposed table structure QA dataset. Their demonstrations effectively showcase TAMO’s impact across multiple LLMs (Llama2, TableLlama, and Mistral) and training approaches (SFT, LoRA).

**Weaknesses:**

1. While the hypergraph structure is theoretically permutation-invariant, incorporating hypergraph embeddings into LLMs may introduce potential errors. Additionally, the paper lacks baselines or a detailed discussion on challenges related to embedding hypergraph structures within LLMs.
2. The paper lacks evaluation/analysis on learned hypergraph representation, such as predicting the table structure from the hypergraph embedding.

**Questions:**

1. Do the authors evaluate the learned hypergraph representation by tasks like predicting the table structure from the hypergraph embedding?
2. What training data was used for TAMO in each experiment? I assume the training set with the evaluation datasets were used. Given that the hypergraph relies on table structure, could TAMO training be conducted solely on synthetic data, such as StructQA questions?

In general, I think it is a good paper. I am open to adjusting my scores if these aspects can be clarified.

---

### Author Response · Authors · 2024-12-03
**Reminder: Discussion Phase Ending Soon – Your Feedback Matters!**

Dear Reviewers,

We hope this message finds you well. We are deeply grateful for your reviews and recognition, which have significantly strengthened our work. With your insights, we believe **TAMO, the first multimodal LLM specifically tailored for the tabular domain**, is paving the way for a significant new research frontier.

***As the discussion phase ends in less than 8 hours***, we’d like to express our thanks individually:

1. **Reviewer `WgPY`**: We extend our heartfelt gratitude for your unwavering support and recognition from the very beginning. **Your strong rating of 8, combined with the highest level of confidence at 5, means a great deal to us.** We are also grateful for your willingness to help convince other reviewers. Your affirmation is a significant encouragement, and we hope your valuable insights will be favorably considered by the ACs and PCs.

2. **Reviewer `BCkd`**: We deeply appreciate your meticulous review and detailed suggestions, which have greatly enhanced our work. Your acknowledgment that **TAMO "may deserve a note of 8"** is encouraging, and we look forward to your latest thoughts, as your feedback is invaluable. We trust that our detailed responses have addressed your concerns.

3. **Reviewers `aQGU` and `Vyb4`**: Thank you for your initial review and **your openness to increasing your ratings based on our revisions**. We have carefully addressed your concerns and hope the updated paper resolves any lingering doubts. It has been several weeks since we last heard from you, and we would greatly appreciate reconnecting before the discussion phase concludes.

TAMO is opening new doors in multimodal LLMs for table modality, and every reviewer’s contribution will play a critical role in shaping this emerging field. Your support at this stage can make a lasting impact. ***Let’s shape this frontier together!***

Thank you again for your time and valuable input. We look forward to hearing your final thoughts.

Best regards,

The Authors of Submission14078

---

> ### Author Response · Authors · 2024-12-03
>
> Dear Reviewers,
>
> We hope this message finds you well. We are deeply grateful for your reviews and recognition, which have significantly strengthened our work. With your insights, we believe **TAMO, the first multimodal LLM specifically tailored for the tabular domain**, is paving the way for a significant new research frontier.
>
> ***As the discussion phase ends in less than `3 hours`***, we’d like to express our thanks individually:
>
> 1. **Reviewer `WgPY`**: We extend our heartfelt gratitude for your unwavering support and recognition from the very beginning. **Your strong rating of 8, combined with the highest level of confidence at 5, means a great deal to us.** We are also grateful for your willingness to help convince other reviewers. Your affirmation is a significant encouragement, and we hope your valuable insights will be favorably considered by the ACs and PCs.
>
> 2. **Reviewer `BCkd`**: We deeply appreciate your meticulous review and detailed suggestions, which have greatly enhanced our work. Your acknowledgment that **TAMO "may deserve a note of 8"** is encouraging, and we look forward to your latest thoughts, as your feedback is invaluable. We trust that our detailed responses have addressed your concerns regarding the clarification of assumption and language improvements.
>
> 3. **Reviewers `aQGU` and `Vyb4`**: Thank you for your initial review and **your openness to increasing your ratings based on our revisions**. We have carefully addressed your concerns and hope the updated paper resolves any lingering doubts. It has been several weeks since we last heard from you, and we would greatly appreciate reconnecting before the discussion phase concludes.
>
> TAMO is opening new doors in multimodal LLMs for table modality, and every reviewer’s contribution will play a critical role in shaping this emerging field. ***Your support at this stage can make a lasting impact. Let’s shape this frontier together!***
>
> Thank you again for your time and valuable input. We look forward to hearing your final thoughts.
>
> Best regards,
>
> The Authors of Submission14078

---

### Meta-Review · Area_Chair_tnJB · 2024-12-23

**Metareview:**

This paper presents TAMO, the first multimodal LLM for table modality. It also presents StructQA, a benchmark to evaluate the skills of LLMs in understanding structured data. TAMO uses hypergraphs to learn tabular representations which can seamlessly integrate with existing LLMs as soft prompts. Experiments on five datasets show that TAMO yields significant improvements with an average relative gain of 42.65%.

**Strengths:**

* The research problem of injecting structured information into LLM representations of tabular data is clearly motivated.
* The idea of using hypergraphs to learn structured tabular representations is novel, although this paper can significantly benefit from further discussions on related works using hypergraphs.
* The proposed TAMO dataset can effectively evaluate LLMs in structured tabular data understanding.
* The authors provide comprehensive experiments on five datasets, showing that TAMO enhances structured tabular understanding.

**Weaknesses**

* There are a few missing experiments, such as evaluation of learned hypergraph representation and generalization to larger tables, which were addressed using the rebuttal period.
* Reviewer BCkd also kindly pointed out many grammatical errors and misuse of certain terms, which were fixed in the revision.

The remaining major issues are:

* Technical writing/presentation (BCkd): “The contributions are occasionally overstated without sufficient evidence”.

* Related to the above point, the current submission lacks discussion and comparison with related works in tabular representations learning (BCkd, aQGU, Vyb4), such as HyTrel (Vyb4). While the authors provide additional discussions in the appendix in the revised version, such critical points should be elaborated and carefully addressed in the main text. The authors shall also consider providing additional experimental results to more directly compare with those related efforts, if possible.

While some reviewers are relatively positive with this submission, I believe this paper would benefit from another round of major revision to clear the concerns regarding related works. Therefore, the decision is reject.

**Additional Comments On Reviewer Discussion:**

Please refer to the meta-review.

---

### Decision · Program_Chairs · 2025-01-22

Reject